# Functionalisation of Silicone by Drug-Embedded Chitosan Nanoparticles for Potential Applications in Otorhinolaryngology

**DOI:** 10.3390/ma12060847

**Published:** 2019-03-13

**Authors:** Urban Ajdnik, Lidija Fras Zemljič, Matej Bračič, Uroš Maver, Olivija Plohl, Janez Rebol

**Affiliations:** 1University of Maribor, Faculty of Mechanical Engineering, Institute for Engineering Materials and Design, Smetanova 17, 2000 Maribor, Slovenia; urban.ajdnik@um.si (U.A.); matej.bracic@um.si (M.B.); olivija.plohl@um.si (O.P.); 2Faculty of Medicine, Institute of Biomedical Sciences, University of Maribor, Taborska ulica 8, 2000 Maribor, Slovenia; uros.maver@um.si; 3University Medical Centre Maribor, Department of Otorhinolaryngology, Cervical and Maxillofacial Surgery, Ljubljanska ulica 5, 2000 Maribor, Slovenia

**Keywords:** silicone, tympanostomy tube, chitosan, nanoparticles, drug delivery, antimicrobial activity

## Abstract

Silicones are widely used medical materials that are also applied for tympanostomy tubes with a trending goal to functionalise the surface of the latter to enhance the healing of ear inflammations and other ear diseases, where such medical care is required. This study focuses on silicone surface treatment with various antimicrobial coatings. Polysaccharide coatings in the form of chitosan nanoparticles alone, or with an embedded drug mixture composed of amoxicillin/clavulanic acid (co-amoxiclav) were prepared and applied onto silicone material. Plasma activation was also used as a pre-treatment for activation of the material’s surface for better adhesion of the coatings. The size of the nanoparticles was measured using the DLS method (Dynamic Light Scattering), stability of the dispersion was determined with zeta potential measurements, whilst the physicochemical properties of functionalised silicone materials were examined using the UV-Vis method (Ultraviolet-Visible Spectroscopy), SEM (Scanning Electron Microscopy), XPS (X-Ray Photoelectron Spectroscopy). Moreover, in vitro drug release testing was used to follow the desorption kinetics and antimicrobial properties were tested by a bacterial cell count reduction assay using the standard gram-positive bacteria *Staphylococcus aureus*. The results show silicone materials as suitable materials for tympanostomy tubes, with the coating developed in this study showing excellent antimicrobial and biofilm inhibition properties. This implies a potential for better healing of ear inflammation, making the newly developed approach for the preparation of functionalised tympanostomy tubes promising for further testing towards clinical applications.

## 1. Introduction

The use of implantable medical devices is a common and indispensable part of medical care for both diagnostic and therapeutic purposes [1,2]. However, implantation of medical devices often leads to the occurrence of difficult-to-treat infections, because of the colonization of their abiotic surfaces by biofilm-growing microorganisms, which are increasingly resistant to antimicrobial therapies [3]. A promising strategy to combat device-related infections is based on anti-infective biomaterials that either repel microbes so they cannot attach to the device surfaces, or kill them in the surrounding contact areas [1,2,4,5].

With the use of different medical devices, it is critical to ensure that the active/functional areas remain free of microbial contamination, and the surface functionalisation of the materials plays a major role. Healing efficiency may be improved through the manipulation of the material surface. For the development of new generations of highly effective medical devices, several requirements need to be fulfilled in regard to the material’s surface that interacts with the biological environment. These include biocompatibility, hydrophilicity/hydrophobicity, mechanical resistance, antimicrobial activity, as well as antioxidant and antifouling (anti-biofilm attachment) properties [1,6]. Smart coatings and surface morphology manipulations are, nowadays, the driving force for creating such active surfaces of medical devices. An active release strategy and advanced natural-bio-based principles are priorities [2].

Amongst the most interesting basic materials for medical devices is silicone [7]. The possibilities for its expanded use in medical applications are bright, considering its already successful utilization in orthopedics, as part of various catheters, drains and shunts, contact lenses, artificial organs, as components in kidney dialysis, heart-bypass machines, blood-oxygenators, and many others [8,9,10].

Silicone is a commonly used material for tympanostomy tubes, whose insertion is a leading surgical procedure performed on children worldwide [11]. A tympanostomy tube is implanted surgically—through an incision in the eardrum (myringotomy)—to allow drainage of fluid from the middle ear in patients with *otitis media* (a group of inflammatory diseases of the middle ear). Despite this, persistent otorrhea (ear drainage) is the most common complication following tympanostomy tube insertion and can lead to tube occlusion and discomfort [6].

General acute otitis media is, commonly, caused by microorganisms, including *Streptococcus pneumonia, Haemophilus influenza* and *Moraxella catarrhalis*, while *Staphylococcus aureus* and *Pseudomonas aeruginosa* are typically implicated microorganisms in ottorhea and are likely to have entered the middle ear via the auditory canal through the tympanostomy tube [12].

Antibiotics have been shown to be effective for children with otitis, where the biofilm has not yet been formed. For children with chronic diseases, the medications do not work well enough. In this case, the more effective procedure is the myringotomy. Furthermore, otitis media with effusion and recurrent acute *otitis media* are common problems in children, with a cumulative incidence of up to 80% by the age of 4 years [12].

In addition, tympanostomy tubes are often used to relieve pathologic ear conditions such as Meniere’s disease and serous *otitis media* [13,14,15,16]. When these tubes are inserted into the ear, complications include tympanic membrane perforation, extrusion, tympanosclerosis, post-tympanostomy otorrhea, and further infections, where the treatment is either a topical antibiotic or steroid treatment, or oral antibiotic [12,14,15].

It has been shown that conventional routes of drug administration, such as oral or parenteral routes, are largely ineffective, mainly due to the blood-labyrinth barrier, which is a highly selective semipermeable border that separates the circulating blood from the brain and extracellular fluid in the central nervous system, and, as such, limits (or even prevents) the contact of drugs in the blood stream with the inner ear. Therefore, local (non-systemic) delivery of drugs can be significantly more efficient, and might even be essential for a successful treatment [17,18].

Since myringotomy still prevails as one of the most common surgical interventions for the pediatric population (also important for other patient populations) [12], a major challenge of current research in this field is to find suitable types of active tympanostomy tubes’ functionalisation. In addition to the mechanical function to allow fluid flow and enabling unhindered ventilation of the ear, the functionalisation needs to provide an antimicrobial activity through an effective and controlled drug delivery. Antimicrobial functionalisation of the tube may indirectly reduce inflammation processes (through neutralization of the infection, which causes inflammation) as well as the incidence of biofilm formation on the surface of these tubes. An efficient drug delivery system would improve healing synergistically. A literature review on this topic revealed that only a very limited number of related studies were performed [18], thus providing a large manoeuvrable space for research and development of novel related solutions, e.g., a smart coating for tympanostomy-silicone tubes acting as an antimicrobial surface with instantaneous drug delivery system as proposed in this study.

Tympanostomy tubes with a phosphoryl choline coating have been tested [19]. During the treatment of the inflammatory process in coated and non-coated tympanostomy tubes, no significant differences were observed in the follow-up of 21 and 24 months. Although their study did not find any statistically significant differences between standard tubes and coated tubes, they indicate that their sample size (n = 70) may not have been large enough to allow efficient assessment of the coating and decide whether it leads to the desired improvement [19].

The human serum albumin (HSA) was also used as a coating on standard tympanostomy tubes of various materials [20,21]. Fibronectin, a typical serum protein, which is one of the most adhesive glycoproteins, was used as a microorganism-blocking agent for tympanostomy. HSA coated tubes inhibited fibronectin binding from 59 to 85%, depending on the type of tube used. The study showed the potential of HSA coating in preventing the binding of pus and other undesired secretions in tympanostomy tubes [20,21].

Tympanostomy tubes from solidified polymer melts (Elvax and polyurethane) and [1] three antimicrobial agents (ciprofloxacin, usnic acid and polyhexamethylene biguanide) for insertion into the tympanostomy membrane (ear drum) were fabricated and tested in terms of drug release kinetics as well as antimicrobial activity against *Pseudomonas aeruginosa*, *Staphylococcus aureus*, *Haemophilus influenzae* and *Streptococcus pneumoniae*. The release kinetic curves revealed two stages of antibiotic discharge, with released drug concentrations being above the minimal inhibitory concentrations (MICs) in the first six days of elution [22].

Moreover, three brominated furanones (furanone-1, furanone-2 and furanone-3) applied to the surface of PVC as a tympanostomy tube material did not seem to be effective against *Staphylococcus aureus* biofilm [23] but one of the three, furanone-3, inhibited *Escherichia coli* biofilm formation [23].

Wang et al. investigated the effectiveness of an organoselenium coating on Donaldson tympanostomy tubes based on silicone material for inhibiting biofilm formation using an in vitro study. *Staphylococcus aureus* and nontypable *Haemophilus influenzae* formed considerable biomass on uncoated tympanostomy tubes, while the organoselenium tubes inhibited biofilm formation drastically, thus showing potential as a long lasting inhibitory agent [24].

Ojano-Dirain et al. determined if biofilm formation on silicone tympanostomy tubes could be prevented by commercially available polyvinylpyrrolidone (PVP) or/and silver oxide coatings. The success of their approach was tested while exposing the prepared coatings to human plasma and to cultured *Pseudomonas aeruginosa* or *Staphylococcus aureus*. Biofilm formation after 4 days was assessed by quantitative bacterial counts and SEM, concluding that PVP and silver coatings reduce *Pseudomonas aeruginosa* biofilm formation (PVP was superior to silver), while combining the PVP and silver coatings does not improve biofilm resistance further [25].

Furanones were also loaded into microparticles [26] and nanoparticles [27]. Specifically, Cheng et al. [27] used the halogenated 4-bromo-5-(bromomethylene)-2(5H)-furanone in combination with biodegradable poly-DL-lactic acid nanoparticles to fabricate a novel antibacterial coating for titanium (Ti) implants. Ti material may also be used for tympanostomy tubes. The mentioned antibacterial coating also exhibited an inhibition against *Staphylococcus aureus* throughout a 60-day study period, which is considered a period long enough to prevent implant-related infections in the early and intermediate stages of device implantation. The antibacterial rate was about 100% in the first 10 days and 90% in the following 20 days [28].

Although researchers have attempted to prevent biofilm formation using tympanostomy tubes coated with various antimicrobial compounds (mostly using antibiotics, synthetic polymers and/or inorganic substances), there is still, currently, no type of tympanostomy tube to which bacteria will not adhere [6]. Furthermore, most of the existing solutions are either fully or partially cytotoxic, due to which they may cause an allergenic reaction by contact during use. Therefore, the use of natural, biodegradable and functional polymers is of high interest for scientific and industrial communities.

Chitosan, a natural amino polysaccharide, has been explored widely for biomedical applications like tissue engineering, gene therapy, wound healing and drug delivery. It is particularly attractive in the form of nanoparticles or nanocapsules, as they can provide some advantages over a chitosan coating alone. These include improved antimicrobial efficiency due to their small size, resulting in a high active surface area to volume ratio, as well as their ability to control the release of active agents, and their contribution to improved mucoadhesive properties [29,30].

In our previous work it was shown that chitosan nanoparticles themselves may be especially attractive as a surface coating for different medical materials such as cellulose fibres, silicone catheters, polyethylene terephthalate (PET) vascular grafts, etc. [10,29,31,32]. Moreover, chitosan nanoparticles were also used successfully as a delivery system for iodine and drugs previously attached onto cellulose medical textiles [33,34].

In this paper, chitosan nanoparticles were prepared by ionic gelation, which is a simple, non-time-consuming preparation procedure that can be performed at mild reaction conditions and is additionally free of any toxic reagents. These nanoparticles were used to encapsulate the co-amoxiclav (CoAM) drug mixture, and their consequent adsorption onto O_2_ plasma-activated tympanostomy-silicone tubes was checked. Plasma activation was used to turn the hydrophobic surface character of the tubes into a more hydrophilic one, thus improving the chitosan nanoparticles’ attachment. In our previous work, it has been already shown and discussed that advanced and environmentally friendly cold O_2_ plasma treatment can be used to enhance the adhesion of chitosan onto inert polymer materials [17,35]. The functionalised silicone material was analysed regarding the surface elemental composition, morphology, coating stability, drug release performance and antimicrobial activity. It has been shown that, either the chitosan-nanoparticle-based coating alone, or as part of a drug delivery system, acts as a promising functional layer for tympanostomy-silicone based tubes.

## 2. Materials and Methods

### 2.1. Materials

Chitosan (low molecular weight) and Sodium tripolyphosphate (TPP, purum p.a., ≥ 98.0%), were purchased from Sigma-Aldrich (Taufkirchen, Germany). Acetic acid was purchased from Honeywell (Seelze, Germany). Co-amoxiclav was obtained from Lek (Ljubljana, Slovenia) and normal saline (0.90% *w*/*v* NaCl) from B. Braun (Melsungen, Germany). A silicone elastomer base and silicone curing agent kit SYLGARD^®^ 184 was purchased from DOW (Wiesbaden, Germany), tryptic soy agar (TSA) plates from Millipore (Wien, Austria) and polyethylene terephthalate mesh from Beti (Metlika, Slovenia). All chemicals and materials were used as received, without any further purification. Ultrapure water (with a resistivity of 18.2 MΩ cm, obtained from Milli-Q, Millipore Corporation, MA, USA) was used throughout the experiments.

### 2.2. Preparation of Chitosan, TPP and CoAM Solutions

An appropriate amount of chitosan was dissolved in Milli-Q water to prepare a 1% (*w*/*v*) solution. The pH of the solution was adjusted to 3.8 initially with acetic acid, followed by constant overnight stirring. Acetic acid was then added to adjust the pH to 3.5 and the total volume of the solution was filled to the required amount.

TPP was suspended in Milli-Q water in order to prepare a 0.2% *w*/*v* solution.

Co-amoxiclav was used as a model drug, composed of amoxicillin (amoxicillin sodium) and clavulanic acid (clavulanate potassium) in a 5:1 *w*/*w* ratio. 20 mL of normal saline was added into the vial with the drug (1000 mg/200 mg), and shaken for a further 10 min.

### 2.3. Preparation of Bare and Loaded Chitosan Nanoparticles

Chitosan nanoparticles were prepared by the ionic gelation technique. Simultaneously 0.2% (*w*/*v*) of TPP solution was added to a fixed volume of 1% (*w*/*v*) chitosan solution, in order to obtain a 5:1 chitosan to TPP weight ratio. This ratio was chosen according to the previously published work, reporting it as an optimal ratio for obtaining the desired antimicrobial activity of nanoparticles’ dispersion [36]. Particles were formed spontaneously under magnetic stirring for 1 h at room temperature. The final pH of the chitosan nanoparticles dispersions was adjusted to 4.0 by the addition of concentrated acetic acid. 

Co-amoxiclav embedded chitosan nanoparticles were prepared as follows. 10 mL of chitosan solution was added to a 50 mL beaker, following constant mixing. Further, 10 mL of TPP was added, and, at the same time, 10 mL of co-amoxiclav previously dissolved in saline. Nanoparticles with encapsulated drug formed spontaneously and the dispersion was stirred for 30 min. 

### 2.4. Preparation of Silicone Material and O_2_ Plasma Treatment

Polydimethylsiloxane (PDMS), also known as dimethylpolysiloxane or dimethicone, which belongs to a group of polymeric organosilicon compounds, was used as a representative silicone material. In a 200 mL plastic crucible, 90 g of a silicone elastomer was weighed, to which 10 g of a silicone curing agent was added. The solution was mixed, followed by pouring the resulting solution into plastic molds. These were dried in a vacuum oven for 24 h at 80 °C. A uniform distribution of the polymer solution was required to ensure that silicone plate samples were as comparable as possible. These thin silicone plates simulated the tympanostomy tubes and due to their dimensional and geometrical characteristics, serve as ideal model platforms, allowing for an easier physicochemical characterisation of functional silicone-based materials.

Silicone plates were cleaned, dried and cut to a required size for further activation with O_2_ microwave plasma in a surfatron mode. The forward power was set to 2 kW (I = 0.3 A). O_2_ pressure in the treatment chamber, made of quartz glass was set to 30 Pa. The samples were exposed to O_2_ plasma treatment for the following times: 1 min, 2 min, 3 min and 5 min, respectively. The experiments were done at the Institute “Jožef Stefan”, Ljubljana, Slovenia.

### 2.5. Application of Chitosan Nanoparticles onto Silicone Material

Dispersions were applied to the top side of the silicone plates using an airbrush (SP-575, Sparmax, Taiwan). The nanoparticles’ dispersions were applied to inactivated and plasma-activated samples. The appropriate pressure in the airbrush tubes for application of the sample was regulated through a connection with a nitrogen cylinder. Functionalised silicone samples were stored in a refrigerator at a temperature of 5 °C, due to the drug’s sensitivity to high temperature. Sample description and notation is given in Table 1.

### 2.6. Characterisation of Dispersions

#### 2.6.1. Evaluation of Hydrodynamic Diameter, PDI and ζ-Potential

The average particle hydrodynamic diameter (d_h_) and polydispersity index (PDI) of the prepared nanoparticle dispersions (CN and CN-CoAM) were determined by dynamic light scattering and through the ζ-potential (ZP) measurement using a Zetasizer Nano ZS (Malvern Instruments, Worcestershire, UK). Samples were injected into disposable cuvettes (DTS0012, Malvern Instruments, UK) for dh and PDI measurements or folded capillary cells (DTS1070, Malvern Instruments, UK) for ZP determination. Prior to analysis, the samples were stirred for 15 min and their pH adjusted to 4 using acetic acid (0.1 M), if necessary.

#### 2.6.2. Drug Encapsulation Efficiency

The drug encapsulation efficiency (EE) was determined indirectly after separation of CN-CoAM nanoparticles from medium, containing non-encapsulated co-amoxiclav using a centrifugation-based technique. The percentage of encapsulation efficiency of co-amoxiclav in the nanoparticles was measured using UV-Vis spectrophotometry (275 nm, Cary 60 UV-Vis spectrophotometer, Agilent Technologies, Santa Clara, CA, USA) and calculated as follows:
(1)$$EE\;\left(\%\right)=100\%\times \frac{C_{0}-C_{s}}{C_{0}},$$
where *C_0_* and *C_s_* are total drug concentration used to prepare the particles and the concentration of co-amoxiclav present in the supernatant after centrifugation, respectively. Standard co-amoxiclav calculation curve was plotted beforehand using Milli-Q water.

### 2.7. Surface Characterisation

#### 2.7.1. X-ray Photoelectron Spectroscopy

In order to assess the surface of the sample (functionalised silicone material), XPS spectra were recorded (PHI TFA XPS Physical Electronics, Chanhassen, MN, USA). The base pressure in the XPS analysis chamber was 10−8 Pa. The samples were excited with X-rays using monochromatic Al Kα1.2 radiation (1486.6 eV) operating at 200 W. Photoelectrons were detected with a hemispherical analyser, positioned at an angle of 45° with respect to the normal to the sample surface. The energy resolution was about 0.6 eV. Spectra were recorded from at least two locations on each sample, using a 400 μm analysis area. Surface elemental composition were calculated from the survey-scan spectra using the Multipak software (ULVAC-PHI, Inc., Chigasaki, Japan)

#### 2.7.2. Scanning Electron Microscopy

Surface morphology was evaluated using scanning electron microscopy (SEM). Prior to imaging, the samples of silicone materials were prepared by cutting the foils into small, approximately 50 mm^2^ × 50 mm^2^ square pieces, which were attached to aluminium sample holders using an adhesive carbon tape to ensure conductivity. Imaging was performed using an SEM (FEI Quanta 200 3D, Hillsboro, OR, USA, whilst the samples were analysed at an accelerating voltage of 1 kV, and at a variable working distance (4–5 mm) using different-sized apertures.

### 2.8. Evaluation of Antimicrobial Activity

To perform microbiological evaluation, the bacterial growth after exposure to respective samples was used to determine the effectiveness of the prepared coated silicone materials. Microbiological testing of the antimicrobial activity of the as-prepared coated silicone plates was carried out using a direct (bacteria attachment to the samples) and indirect (exposure of the bacteria culture to sample extracts) method. For this purpose, a *Staphylococcus aureus* test strain (DSM 799) was used, where the tests were performed according to the internal protocols of the Department of Microbiological Research, Centre for Medical Microbiology of the National Laboratory for Health, Environment and Food in Maribor, i.e. No. P96 Biofilm production on various materials—*Staphylococcus aureus*’. Briefly, sample antimicrobial activity was tested by determining the number of colony-forming units (CFU) in the supernatant after exposure to respective samples. Variously functionalised silicone materials in the size of 10 mm × 10 mm were exposed to the standardized medium inoculated with *Staphylococcus aureus* and adjusted to 0.5 on the McFarland scale [37]. After 4 h of incubation at a temperature of 37 °C and slight shaking, the obtained sample was diluted appropriately and inoculated to a TSA plate. Colonies were counted after incubation on agar plates at appropriate conditions after 24 h and after one month. The effect of the functionalised silicone material was determined as a reduction in growth, where the number of bacteria after exposure to a non-functionalised silicone material was compared to the functionalised one.

### 2.9. In Vitro Drug Release Testing

In vitro drug release studies were performed using an Automated Transdermal Diffusion Cells Sampling System (Logan System 912-6, Somerset, KY, USA). The drug-loaded samples PDMS_CN-CoAM_, PDMS_PA1, CN-CoAM_, PDMS_PA2,CN-CoAM_, PDMS_PA3, CN-CoAM_, PDMS_PA5, CN-CoAM_ were cut into 10 mm × 10 mm squares and placed on the top of a PET mesh. The receptor compartment was filled with Phosphate Buffered Saline (PBS, purchased from Sigma-Aldrich, Germany) with a pH value of 7.4 and its temperature was maintained at 37 °C. During the dissolution testing the medium was stirred continuously with a magnetic bar. Samples were collected over a period of 24 h at different time intervals (1, 5, 10, 20, 30, 60, 120, 180, 240, 300, 360 and 1440 min), while the released/dissolved CoAM concentration in the receptor medium was determined by a UV-Vis spectrophotometer (Cary 60 UV-Visible Spectrophotometer, Agilent, Germany) by quantification of the absorption band at 276 nm. The withdrawn sample volumes were replaced by fresh PBS with a stable temperature of 37 °C of the same volume. Sink conditions were assured due to sample withdrawal, followed by sample dilution through media replacement. In calculation of concentrations using the Beer–Lambert Law, this dilution was accounted for. All release studies were performed in triplicates and are reported as average value with standard errors.

## 3. Results and Discussion

### 3.1. Nanoparticle Dispersions

#### 3.1.1. Hydrodynamic Diameter, PDI and ζ-Potential

Initial d_h_ and PDI measurements were performed using blank CN, since we wanted to evaluate if it is necessary to use an ultrasonic bath (UB) to prepare a stable nanoparticle suspension. The obtained data for the particles prior to UB (Table 2), resulted in a larger particle d_h_ (at approximately 1470 nm) and a PDI of 0.52 nm, indicating medium particle heterogeneity. Ultrasonic treatment led to reduction of the particle dh. Since sedimentation of smaller particles is slower, these are more favourable to prepare stable suspensions. PDI after the UB treatment shows elevated values, pointing to a higher polydispersity of particles. Average CN after UB particles d_h_ triplicate shows 379.70 nm.

Following the preparation of drug-loaded nanoparticles (CN-CoAM), and considering the favourable effect of UB, these were also exposed to UB treatment and analysed. A roughly 35% increase of d_h_ from 379.90 nm to 514.43 nm in comparison to UB treated CN indicated drug encapsulation into the particle’s interior. Based on the measurements (and considering results from other methods, proving successful drug incorporation—see below), it can be assumed that drug encapsulation induced particles enlargement, while PDI lowered, showing lower particle d_h_ heterogeneity (0.48).

Next, we compared the influence of the UB and drug encapsulation on the sample ZP (Table 2). Generally, the obtained ZP values indicate particle stability if values are in the range over +30 mV or less than −30 mV; low values, between −5 mV and +5 mV can lead to agglomeration [38]. The first observation based on the ZP results is that expectedly, the UB does not change the ZP of the sample by a large margin (comparison between CN before and after the UB). Considering the absolute value of the measured ZP for these two samples (ZP ~ 30 mV) also indicated their stability, according to the above-mentioned literature source. In comparison, the particles in the CN-CoAM sample seem to be the least stable ones (ZP ~ 5.40 mV). Whereas the CN particle stability is expected, due to the protonated amino groups of chitosan in acidic medium, and, consequently, sufficient repulsive forces due to protonated amino groups, this evidently does not happen in the case of the CN-CoAM. Considering the results (ZP measurements, as well as the increased d_h_ values) and the molecular structure of co-amoxiclav (which is composed of two compounds with pKa values in the acidic region) has likely bound to particles’ surfaces, causing the lowering of the positive (amino functional group related) charge, by limiting the available chitosan amino groups. To a lesser extent, but also possible, co-amoxiclav could also bind to amino groups chemically during the encapsulation. Agglomeration of CN particles dispersion is not expected, while CN-CoAM tend to agglomerate, as the obtained d_h_ and ZP results show particle agglomeration and sedimentation.

#### 3.1.2. Drug Encapsulation Efficiency

Based on the results from ZP and DLS measurements, it has been concluded that the particles formed successfully, and therefore the next step was to evaluate how much of the co-amoxiclav was actually incorporated into the chitosan-loaded particles. For this purpose, we evaluated the co-amoxiclav encapsulation efficiency (EE) in accordance with Equation (1), which was found to be approximately 91.62%, which may be considered as very effective drug loading. EE depends on the drug used, the applied loading method, as well as the conditions used to encapsulate the active substance. This result gave as further confirmation that the used TPP:chitosan ratio yields particles suitable for preparation of drug-loaded chitosan-based formulations.

There are already some studies reporting the use of differently prepared chitosan-based particles for drug encapsulation, whereas their EE using Equation (1) varies a lot. For example, Ristić et al. [33] showed an 87% EE of clindamycin 2-phospate incorporated into chitosan nanoparticles. Previously, tenofovir in a mixture of water and ethanol rendered EE of around 6%, whereas their optimal conditions for drug encapsulation, led to an EE of 20% [29]. Doxorubicin was also encapsulated into chitosan nanoparticles with a 68% EE by using the electrospray ionization method [39,40]. This is another confirmation that our proposed formulation led to a very high drug EE.

### 3.2. Functionalised Silicone

#### 3.2.1. Surface Analysis by XPS

Binding energy peaks of elements of inactivated silicone (top left), CN coated silicone (top right), and CN-CoAM coated silicone (bottom) are represented in Figure 1.

From Figure 1 one can observe that, in the case of inactivated silicone, the highest peak comes from 1s hybridised carbon followed by 1s hybridised oxygen, which represents a third of all the elements. This is followed by 2p-hybridised silicon. In the case of the CN coated silicone the 1s hybridised carbon and oxygen are the predominant ones as in the case of the neat silicone, followed by 2p hybridised silicon. However, a peak for nitrogen (0.6 at.%) can also be observed, which indicates that chitosan is present on the CN coated silicone surface. In the case of the CN-CoAM, an increase in carbon (54.6 at.%) and a decrease in silicone (16 at.%) can be observed in comparison to the neat silicone surface indicating the presence of the coating on the silicone surface. The presence of nitrogen confirms that the chitosan nanoparticles with embedded drug were bound onto PDMS films. Besides that, small amounts (below 2 at.%) of chlorine, sodium, and phosphorus were detected as well. These are integral parts of the co-amoxiclav active ingredient and confirm its presence in the CN coating. The atomic percentages of all atoms present in the measured samples are shown in Table 3.

Table 3 shows the detailed elemental analysis of the inactivated and coated inactivated silicone samples, calculated from the binding energy peaks. The quantitative amount of each element is shown in atomic percentage (at.%). It can be seen that the neat and inactivated silicone contains 47.4 at.% of C, 28.8 at.% of O_2_, and 24.1 at.% of Si. This is in good agreement with the theory, as the chemical structure of PDMS contains 50 at.% of C, and 25 at.% of both O_2_ and Si. In the case of the CN coating onto PDMS, one can observe insignificant changes in the at.% of C, O_2_, and Si. A low amount of N (0.6 at.%) is present in this sample, which indicates that a very thin or inhomogeneous nanoparticle coating is formed on the silicone surface. In the case of the CN-CoAM coated PDMS one can observe a significant decrease in the Si at.% from the initial 24.1 at.% to 16.0 at.%, which shows that the applied coating is denser than in the previous case, as less of the Si from the silicone surface is present in the overall surface evaluation. Besides that, the C at.% increased to 55.2 at.%, and new elements from the co-amoxiclav are present in the surface, such as Cl (1.5 at.%), S (0.3 at.%), and Na (1.0 at.%). A low amount of P (0.4 at.%) from the TPP, can also be observed in this sample. Furthermore, the amount of N increased to 1.5 at.% and this is another indication of chitosan and co-amoxiclav presence in this coating. The co-amoxiclav consists of amoxicillin (Figure 1), which also contains nitrogen. The combined results from Figure 1 and Table 3 prove the presence of chitosan nanoparticles, as well as chitosan nanoparticles with embedded co-amoxiclav on the PDMS surface.

The activation of silicone with O_2_ plasma, leading to formation of new oxygen-containing functionalities, increases its surface free energy, which in turn allows for a higher coating efficiency with the nanoparticle coatings. Therefore, it is expected that the adsorption of chitosan nanoparticles would be higher, as well as more resistant. Activation with O_2_ plasma was carried out at different exposure times (1 min, 2 min, 3 min and 5 min). It has been shown that the silicone surfaces exposed to O_2_ exhibit the same primary peaks of elements as the inactivated sample, namely C, O_2_, and Si, regardless of the exposure time. The at.% on the other hand, changed significantly, as the oxidation of the surface produced new oxygen-containing functionalities. Therefore, the plasma-activated samples contain higher at.% of oxygen (Figure 2). Among all plasma-activated samples, PDMS_PA5_ showed the highest increase of oxygen-containing groups (Figure 2, Table 4).

This sample contains 55.8 at.% of O, and only 13.9 at.% of C. Based on these results, we presume that the adsorption efficiency of chitosan nanoparticles, alone or the ones with the embedded drug, will be best in the case of this sample; PDMS_PA5_. 

The latter may be seen more clearly from Table 4, which shows surface elemental compositions (in at.%) of the plasma-activated silicone surfaces (1 min, 2 min, 3 min and 5 min) coated either by chitosan nanoparticles alone, or in with the ones loaded with the drug. It has been shown that the exposure time to plasma activation has some influence on the binding between the activated silicone surface and the chitosan-nanoparticle coatings alone or with embedded drug. Increasing the exposure time increased the amount of oxygen-containing groups, which led to a higher deposition of respective coatings on the surfaces (better adsorption). This can be observed by the increase of at.% of C from 49.1 at.%, 56.2 at.% in the CN samples when the plasma- exposure time increased from 1 min to 5 min. The same phenomenon can be observed for the at.% of N (4.0 at.% to 5.1 at.%). It has to be pointed out that a much higher content of nitrogen is detected for these samples in comparison with plasma-inactivated samples, where, for example, in the case of PDMS_CN_, only 0.6. at.% of nitrogen was detected. An opposite trend is seen for the O and Si atoms, which are decreasing gradually as more coating is applied to the activated silicone surface.

The same trend is also observed in the case of the CN-CoAM coatings on plasma-activated samples. Additionally for these samples, a much higher amount of nitrogen was detected for activated samples in comparison with plasma inactivated (that were otherwise treated with the coating using the same procedures). This again indicates an improved chitosan-nanoparticle coating adsorption onto PDMS, if the latter is pre-treated by plasma activation. When plasma-activated silicone surfaces (1 min, 2 min, 3 min and 5 min) coated by chitosan nanoparticles alone are compared to those coated by chitosan nanoparticles with embedded drug, it is seen clearly that the latter possess a bit higher nitrogen content that is attributed to the presence of the drug (N is in the drug structure as well).

Moreover, a clear increase in S is observed with increasing plasma exposure times, which is a clear indicator of the increasing amount of the co-amoxiclav drug present in the coatings (Table 4). From these results, one can conclude that activating the silicone surface with O_2_ plasma prior to interactions with the nanoparticles coatings, increases the adsorbed amount of the coatings by up to 10-times when compared to the inactivated samples. Furthermore, as already pointed out, the exposure time also plays a crucial role. Namely, increasing the O_2_ plasma contact time results in an increase of the adsorbed amount of surface coatings.

#### 3.2.2. SEM Micrographs Morphology Evaluation

SEM was used to take a closer look and to analyse the morphology of the untreated PDMS films with respect to modified ones, (e.g., inactivated/activated plasma treated-PDMS, as well as the ones with the deposited formulations). Figure 3 shows the SEM micrographs for the most representative respective samples. Figure 3 shows only the micrographs of samples that were plasma treated for 5 min (PDMS_PA5,CN_ and PDMS_PA5,CN-CoAM_), which was found to present the optimal treatment (plasma activation) time (discussed above).

The main focus of morphology analysis was the evaluation of respective chitosan-nanoparticle coating adhesion to the plasma-activated/inactivated PDMS surfaces. Furthermore, the homogeneity of the formed deposits, together with self-assembled structures that can be formed onto substrate, were observed additionally. The structure of the unmodified (i.e., pristine) PDMS film is shown in Figure 3a. The morphology of the film is mainly smooth and flat with the presence of some small particles that are most likely contaminants, originating from the SEM sample preparation. Furthermore, this sample serves as a reference, in order to see potential differences more clearly after deposition of chitosan-based coatings alone, or with embedded drug onto inactivated/O_2_ plasma-activated PDMS films.

The deposition of the CN formulation (sample PDMS_CN_) onto the PDMS films was clearly successful, as can be seen from Figure 3b. However, due to the difference in the polarity of the PDMS substrate and CN nanoparticles, the latter deposited mainly in the form of agglomerates by minimizing its surface energy [41]. Moreover, the manner of the CN dispersion application with the use of the airbrush spray (spraying technology) may, in part, also contribute to the inhomogeneous distribution of the formulation. Nevertheless, despite all of the above mentioned, CN nanoparticles in the size range of about 500 nm (aligned into lines) can also be observed (Figure 3b; inset). In agreement with the PDI as a measure of distribution, the nanoparticles express large size distribution, and the latter can evidently be seen from the SEM images as well (Figure 3b).

In contrast to the deposed CN formulation described above, a very different morphology of the CN formulation was observed with embedded drug (PDMS_CN-CoAM_) (Figure 3c). Although the applied dispersion covered the PDMS film more uniformly, individual particles are barely observed, as they were embedded into the excess of the polymer and/or drug. It should be noted that the white edge appeared due to the charging of the nonconductive materials. The sputtering of the polymer materials in this case was avoided, because it can potentially influence the already formed morphology (as well as it could hide important morphological features). The white edge also implies the boundary of the formulation’s coverage and the non-covered part of the PDMS film, which again suggests that, a non-homogenous adsorption. 

In order to achieve a more polar film surface, as well as to form surface functional groups on the PDMS surface, which would serve as binding sites for chitosan nanoparticles, the latter was exposed to the O_2_ plasma activation. As mentioned above, after these (pre)activation steps, the formed CN with/without co-amoxiclav were applied onto these films (Figure 3d,e show the samples PDMS_PA5, CN_ and PDMS_PA5, CN-CoAM_, respectively). These samples were chosen, since (according to other analysis methods) the 5 min plasma-treatment showed the most positive influence on the binding of chitosan nanoparticles alone, as well as the ones with embedded co-amoxiclav. To analyse the effect of drug encapsulation on coating deposition in relation to treatment with O_2_ plasma, two different samples were compared (c and d). In general, after plasma activation the adhesion of the applied formulation was improved significantly, which indicates the more hydrophilic character of the activated PDMS film, and better adhesion of the formulation. Moreover, no more large agglomerates were observed despite a wide particle size distribution. As seen from Figure 3, the morphology of the adsorbed particles varies with adsorption of different formulations. In fact, PDMS_PA5,CN_ showed a particulate surface morphology, exhibiting the particles aligned into spherical structures (Figure 3d). The latter might indicate that smaller particles were somehow attracted to each other, maybe even in part into bigger “bubbles”, consisting of separate smaller particles inside, which burst during vacuumization prior to SEM. On the other hand, the PDMS films treated and modified with CN-CoAM show not only a more uniform surface coverage with the particles, but also a much more homogenous observable particle size on the plasma treated-PDMS film, forming interested self-assembled structures (Figure 3e). Regarding homogeneity, the application of CN formulation onto plasma-activated PDMS films did not result in a significant difference with respect to deposed CN on in-activated PDMS films (Figure 3b,c). On the contrary, when CN-CoAM is deposited onto activated PDMS films, this results in a more uniform and homogeneously covered film of applied formulations onto the substrate (Figure 3c,e).

### 3.3. Antimicrobial Assay

The antimicrobial properties of functionalised silicone samples were tested against *Staphylococcus aureus* (DSM 799), shown in Table 5. The reduction is expressed in % reduction according to control (silicone only). The analysis took place the day after the application of the coating, and after one-month of exposure, to check the coating efficiency in the case of prolonged use, as well as to evaluate the time-dependent antimicrobial activity of the coatings in general. The most favorable results were found in a silicone inactivated carrier coated by chitosan nanoparticles with encapsulated drug (bacterial growth decreased by as much as 93, 98%), as well as in the pre-activated O_2_ plasma sample (t = 5 min) with absorbed chitosan nanoparticles with embedded drug. After one month, the efficiency against *Staphylococcus aureus* of the PDMS_CN-CoAM_ sample was still very favourable (99.75%), while the effectiveness of the PDMS_PA5,CN-CoAM_ was considerably reduced, but still quite efficient (a decrease from 97.5% to 67.37%). The latter can be connected to potential changes in the polymer conformation, the partial degradation of the drug-loaded chitosan coating, and, finally, because of the potential degradation of the incorporated antibiotic itself.

From these results, it may be seen that the encapsulated drug improves the antimicrobial properties.

The reduction of bacteria fixation on the surface of silicone samples (direct method) at time 1 month and after 4 h of exposure (CFU/mL after sonication) with *Staphylococcus aureus* DSM 799 was also checked (Table 6).

The test was performed one month after application of the coating, thus testing the stability of samples. The reduction is expressed in percentages according to control (silicone only). The best results were obtained in the case of PDMS_PA5, CN-CoAM_ (drug-loaded chitosan nanoparticles on 5 min plasma-activated PDMS films). Growth in this case decreased by as much as 96.58%. Similarly, favourable results were obtained for the coating that was plasma activated for 1 min (PDMS_PA1, CN-CoAM)_, where the growth of *Staphylococcus aureus* decreased by 95.12%. Good results were also obtained for the sample with the same composition that was not treated with O_2_ plasma (PDMS_CN-CoAM_), where bacterial growth was reduced by 93.05%. These results therefore indicate an excellent efficiency of the coating in prevention of biofilm formation on the tube surface.

The samples previously activated with plasma by 1 min were also analysed, in order to see the influence of plasma activation time on final functional /nanoparticles by embedded drug/silicone microbial properties. As pointed out above, the prolongation of plasma time increased the adsorption of CN as well as CN-CoAM, onto silicone. In general, the antimicrobial efficiency was not influenced significantly with this increase of activation time. The obtained results imply that the antimicrobial activities of silicones functionalised by chitosan nanoparticles alone or with the embedded drug are influenced by a number of factors, and no single mode of action can be determined as the prevailing one. Besides amino groups’ content, the adsorbed polymer conformation, morphology of coatings, drug release profile (discussed below) and accessibility to biological environment may play important roles [31].

### 3.4. In Vitro Drug Release Testing

The measured release data are presented in the form of three different graphs, in which we show the released drug concentration, and cumulative released mass, and the percentage of the release drug as a function of time (Figure 4, Figure 5 and Figure 6). It has to be stated that the latter two types of data representation already account for dilution, which occurs due to sample removal, followed by replacement by fresh media. Therefore, these data can be discussed directly. Some more caution is necessary in reviewing and discussion of the obtained release drug concentration profiles. The dilution was not accounted for in this case, since we wanted to evaluate possible trends in the release profiles preferentially, which might give us clues about more complex release mechanisms that govern the overall drug release and might have an important impact on the overall drug action (in this case on the antimicrobial activity of co-amoxiclav). Furthermore, drops or sudden jumps in concentration can indicate specific sample changes (e.g., degradation or layer removal in case of the multi-layered samples), and could also present clues about the potential sample transformations that also affect the other properties of the prepared samples (e.g., in the case of the observed measurements of antimicrobial activity in the previous chapter).

The release data, regardless of the representation type, are shown until 1440 min. After the therapeutic drug concentration is reached, a constant release with a “stable” drug concentration over time is desired in the targeted application (e.g., in-ear treatment of bacterial infections). Namely, to ensure an effective therapy the bacterial cells need to be exposed to a constant antibiotic concentration in the therapeutic window for the time of exposure to a single tympanostomy tube (after this, the latter needs to be exchanged). Through this, also the body’s own immune response can be boosted to some degree. However, one has also to take into account potential drawbacks of use of antibiotics, which in some cases even lower the response of the body’s immune system. The latter is especially true in use of broad-spectrum antibiotics, which affect also the healthy commensal microbiome that has often a positive contribution to the immune response against pathogens. Considering the observed profiles shown in Figure 4, all samples exhibit the profile, desired in the targeted therapy, in which first a high concentration increase is observed (which enables a fast-achieved therapeutic concentration), followed by a more or less stable release, ensuring a high enough concentration for 24 h (more discussion about this is included below in consideration of the other release data representation types). The least changes (or “pulses”) in the release profile, were found for the sample PDMS_PA5,CN-CoAM_. In vitro release studies showed that the highest release drug concentrations could be achieved in the case of the PDMS_PA1, CN-CoAM_ (i.e., the plasma-activated silicone sample that was plasma treated for 1 min). This sample shows a quite good *Staphylococcus aureus* growth reduction measured as the diminished bacterial fixation on the surface of different silicone samples. For the second-best sample in regard to bacteria growth reduction (PDMS_PA5_, _CN-CoAM_) a similar release profile to the above-mentioned one was achieved, showing a trend relating the adsorption of the chitosan/drug formulation, the release profile, as well as the therapeutic potential (antimicrobial activity) in an expected manner.

Observed initially high drug concentrations are most likely related to the diffusion coefficient between the release media and the drug (and/or the surface layer in touch with the media), which seems to favour the release. In the case of the sample that was not plasma treated, we can see that the incorporated drug amount at the start of the testing is the highest, most likely indicating that the surface coating, containing the drug, does not form strong interactions with the silicone substrate. This might, in turn, lead to less repeatable drug release results, possibly even leading to drug concentrations outside the desired therapeutic window.

The next type of data representation is shown in Figure 5, which shows the cumulative released drug mass as a function of time. In this, we again see that the highest overall release mass can be achieved by the untreated sample (at 360 min), whereas the lowest release concentration at the same time was measured for the sample that was plasma treated for the longest period (5 min). The first important observation we can make by just looking at the general release profiles in Figure 5, is that the release amount (and hence the patient therapy) can be tailored to the needs of an individual patient by adjusting the time of plasma-treatment. Furthermore, this not only allows us to tailor the release drug dose, but also, considering the different inclinations of the respective release profiles, also the release rate. This might even allow the physician to prolong a certain treatment using the same dose (e.g., in the case of specific patient diagnoses, where a slower release can be beneficial for the treatment outcome). A further evaluation of the release profiles shown in Figure 5 leads to another important conclusion, namely, that the initial couple of minutes (until the c_max_ is reached) seems to follow the same fast release rate for all samples. This has another clinical implication. Namely, regardless of the incorporated drug amount, the initially high concentration is reached in all cases, hence allowing for an immediate start of the therapy after almost the same time (onset is almost the same for all the samples). After 400 min of release, all samples reached a more stable (even a plateau like) release. This part can be used as the maintenance phase of the therapy (this part is also in agreement with the discussed release results for Figure 4).

Considering now the final released drug masses, we can see that a 1.8–2.0 mg/mL of the incorporated drug could be achieved. Considering now the volume of the used Franz diffusion cells in the release testing (15 mL), we can calculate the final release drug amount in the range of 27.0–30.0 mg. Since the initial incorporated drug amount was 45.81 mg, we can now also calculate also the efficiency of the drug release from the prepared samples (in 24 h). The latter is between 59–66% (as can be easily seen from Figure 6, which show the % of the release drug), which is good in regard to the potential clinical application of such materials, where the tympanostomy tubes stay in the ear for longer than 24 h. Here, it also has to be stated that the release testing was performed in a “slightly” alkaline medium (7.4 pH), at which the used chitosan nanoparticles are not soluble, which might have affected the release performance further. In the ear, the pH is slightly acidic, which could boost the release further. The latter was also confirmed through an additional experiment for the sample PDMS_PA5, CN-CoAM_, chosen as an optimal sample from antimicrobial point of view, in which we prepared a medium with a pH of 6.0, where a higher drug concentration was achieved at all-time points of the release testing. In general, the 20–30% of higher release was obtained, mostly due to the better solubility of chitosan.

The lower final release in % might be also connected with possible interactions of the chitosan nanoparticles and the coating. However, this aspect should be studied in more detail in a separate study.

The final representation of the data shows the % of the release drug amount over time (Figure 6A). Such a representation is commonly applied to evaluate, if there are any changes in the release mechanism between samples containing the same drug, but in different amounts. From Figure 6 we can clearly see that the release profiles exhibit the same shape of the curves, which confirms a further important clinical implication for the samples, namely that the drug dose can be easily adjusted to patient-specific needs without changing the general release mechanism that could potentially affect the therapy. Finally, we wanted to get more insight about the actual release mechanism. For this purpose, we initially performed the calculation of the first derivatives based on the obtained data, which can expose potential changes in the release mechanism during the time of the dissolution testing [42,43]. From Figure 6B we might conclude that the release is most likely combined from different types of mechanisms. Based on the sample composition, these comprises diffusion- and erosion-based processes, as well as potentially some additional swelling. Due to the latter, commonly employed models to fit the data do not seem to be suitable to be used in this case. Nevertheless, we have conducted some additional fitting with two such models, namely the Weibull model (Appendix A, Appendix A) and the Korsmayer–Peppas model (Appendix A, Appendix A). The data is shown in the Appendix A.

In general, the obtained results are in agreement with the results of the measured antimicrobial activity, especially in the case of the samples that were plasma activated for 1 min and 5 min, respectively. In the latter, case, where lower cumulative masses were measured in the timeframe of 24 h, a prolonged release testing might have revealed that the whole incorporated drug amount was released later. This is the more likely, considering the higher inclination of the plateau part of the release profile for this sample (Figure 5).

## 4. Conclusions

The purpose of the presented research was the preparation of a suitable nano-coating for tympanostomy tubes and their potential for medical use. The polysaccharide coatings, based on chitosan nanoparticles alone, or with the embedded model drug co-amoxiclav were prepared and adsorbed onto silicone tubes. Plasma activation was also used as a pre-treatment for activation of the material for a potential improved adhesion of coatings, and, hence, effectiveness for the desired purpose. The effectiveness of the encapsulation efficiency of the drug into nanoparticles was analysed by the UV-Vis method and SEM. The results were very favourable, as more than 90% of co-amoxiclav could be loaded (encapsulation efficiency) into the particles, indicating that the prepared system presents an excellent drug delivery system.

XPS and SEM methods have proven that the adsorption of all chitosan-nanoparticle systems onto silicone material was successful. It has been further shown that increasing the O_2_ plasma activation time led to an increase of the adsorbed amount of surface coatings by up to 10-times when compared to the inactivated samples. Moreover, through plasma activation, a uniform and homogeneous film coating of the applied formulations was achieved on the silicone-based substrates.

It has been found that the results of reduction of bacterial adhesion on the surface of different silicone samples were better (a higher percentage of decreased growth of microorganisms) with longer use of oxygen plasma prior to the adsorption of chitosan nanoparticles alone or with the embedded drug. The suitability of the drug release from the prepared formulations was checked using the Franz cell method, which was in agreement with the results obtained by other methods. Altogether, the results have shown that the prepared formulations have a potential for future tympanostomy tubes applications, pending further testing in clinical settings.

## Figures and Tables

**Figure 1 materials-12-00847-f001:**
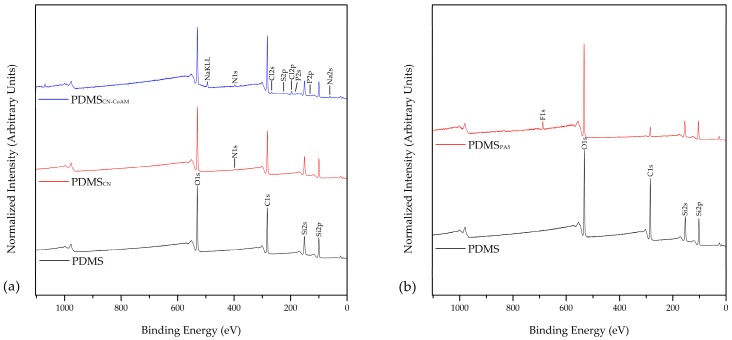
Binding energy peaks of elements of neat inactivated silicone PDMS, PDMS_CN_ and PDMS_CN-CoAM_ (**a**). Binding energy peaks of elements of PDMS and O_2_ plasma (PDMS_PA5_) activated silicone (**b**).

**Figure 2 materials-12-00847-f002:**
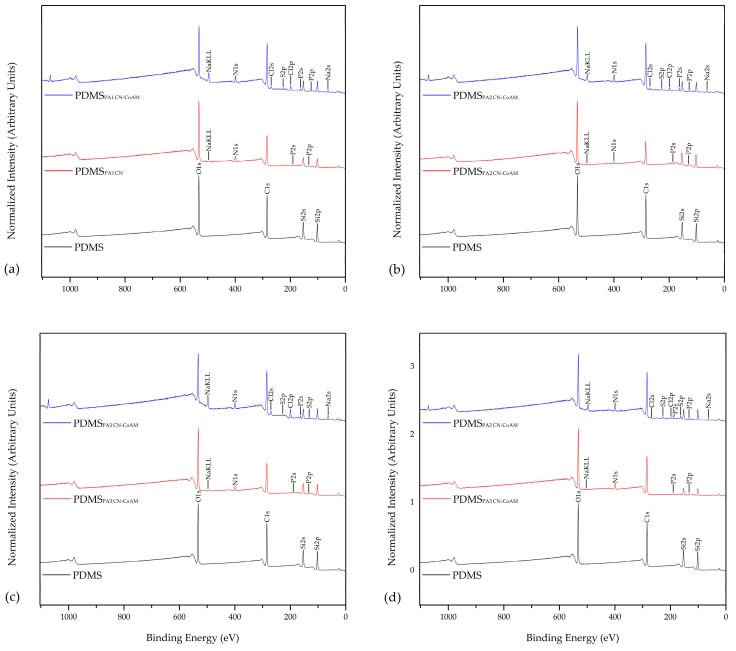
XPS analyses: Binding energy peaks of elements of O_2_ plasma-activated silicone for 1 min (**a**), 2 min (**b**), 3 min (**c**) and 5 min (**d**), coated by chitosan nanoparticles alone or in combination with the drug.

**Figure 3 materials-12-00847-f003:**
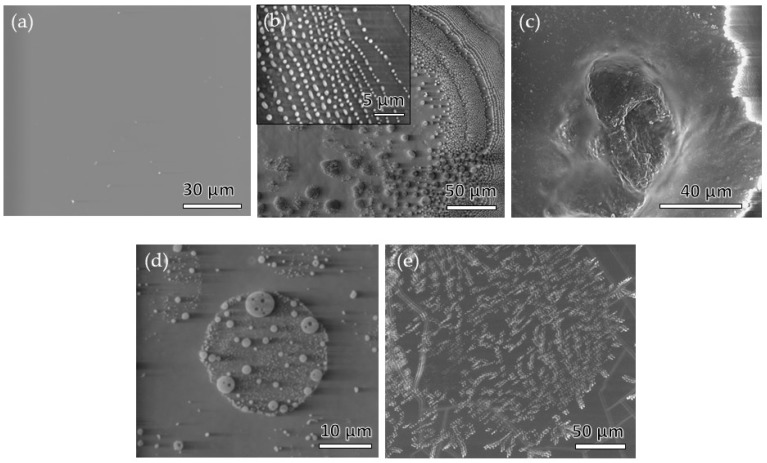
SEM images of the bare PDMS film (**a**), PDMS_CN_ (**b**) and PDMS_CN-CoAM_ (**c**), PDMS_PA5, CN_ (**d**) and PDMS_PA5,CN-CoAM_ (**e**).

**Figure 4 materials-12-00847-f004:**
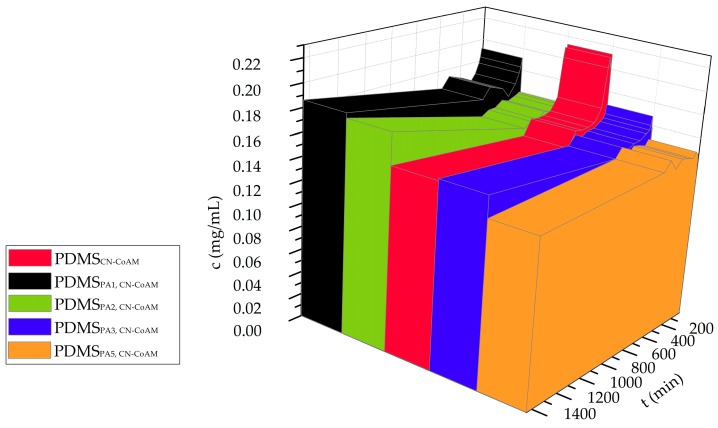
Time-dependent change in the active substance concentration.

**Figure 5 materials-12-00847-f005:**
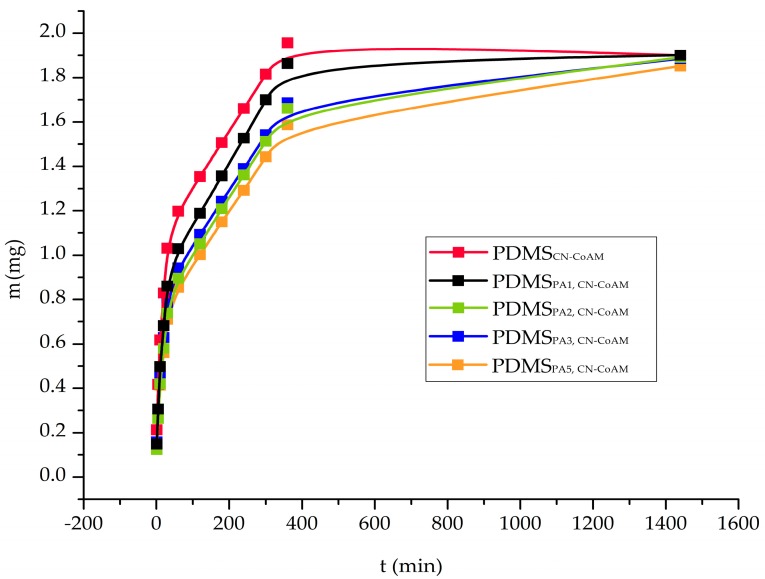
Time-dependent change in the cumulative released mass of the incorporated drug.

**Figure 6 materials-12-00847-f006:**
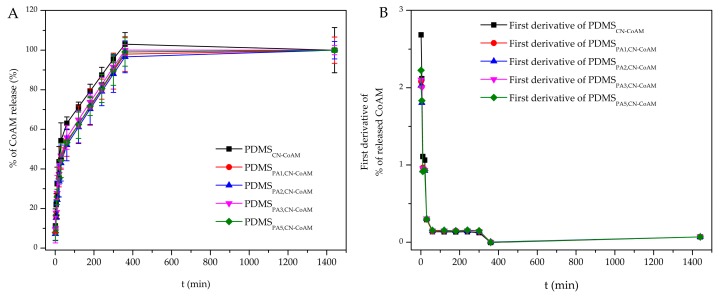
Time-dependent change in the percentage of the released incorporated drug.

**Table 1 materials-12-00847-t001:** List of abbreviations and their descriptions. Samples notation is given in the same table.

**Abbrevation**	**Full Name/Description**
CFU	Colony-forming units
CoAM	Co-amoxiclav
DLS	Dynamic light scattering
EE	Encapsulation efficiency
HSA	Human serum albumin
MIC	Minimal inhibitory concentration
PBS	Phosphate buffered saline
PDI	Polydispersity index
PDMS	Polydimethylsiloxane
PET	Polyethylene terephthalate
PVP	Polyvinylpyrrolidone
SEM	Scanning electron microscope
TPP	Sodium tripolyphosphate
TSA	Tryptic soy agar
UB	Ultrasonic bath
UV-Vis	Ultraviolet-visible
XPS	X-ray photoelectron spectroscopy
ZP	ζ-potential
	**Notation of Samples**
CN	Chitosan and TPP nanoparticles (CN)
CN-CoAM	CN with encapsulated CoAM (CN-CoAM)
PDMS_CN_	PDMS carrier, CN coating
PDMS_CN-CoAM_	PDMS carrier, CN-CoAM coating
PDMS_PA1, CN_	PDMS carrier, O_2_ plasma activated for 1 min (PA1), CN coating
PDMS_PA1,CN-CoAM_	PDMS carrier, O_2_ plasma activated for 1 min (PA1), CN-CoAM coating
PDMS_PA2,CN_	PDMS carrier, O_2_ plasma activated for 2 min (PA2), CN coating
PDMS_PA2,CN-CoAM_	PDMS carrier, O_2_ plasma activated for 2 min (PA2), CN-CoAM coating
PDMS_PA3, CN_	PDMS carrier, O_2_ plasma activated for 3 min (PA3), CN coating
PDMS_PA3, CN-CoAM_	PDMS carrier, O_2_ plasma activated for 3 min (PA3), CN-CoAM coating
PDMS_PA5_	PDMS carrier, O_2_ plasma activated for 5 min (PA5)
PDMS_PA5, CN-CoAM_	PDMS carrier, O_2_ plasma activated for 5 min (PA5), CN-CoAM coating

**Table 2 materials-12-00847-t002:** d_h_, ZP and PDI of CN particles before ultrasonic bath along with CN and CN-CoAM particles after ultrasonic bath.

	CN before UB	CN after UB	CN-CoAM after UB
d_h_ (nm)	1470.53	379.70	514.43
ζ (mV)	27.00	32.43	5.40
PDI	0.52	1.00	0.48

**Table 3 materials-12-00847-t003:** Elemental analysis of the coated and inactivated silicone surfaces by XPS (the surface depth ≈ 10 nm).

Sample	Atomic Percentage of Elements (at.%) *							
	C	N	O	Na	Si	P	S	Cl
PDMS	47.4	-	28.8	-	24.1	-	-	-
PDMS_CN_	46.4	0.6	29.2	-	23.8	-	-	-
PDMS_CN-CoAM_	55.2	1.4	24.3	1.0	16.0	0.4	0.3	1.5

* The standard deviation was within the range 1–3%.

**Table 4 materials-12-00847-t004:** Elemental analysis of the plasma-activated silicone surfaces (1 min, 2 min, 3 min and 5 min) coated by chitosan nanoparticles alone, or in combination with the drug by XPS.

Sample	Atomic Percentage of Elements (at.%)						
	C	N	O	Na	Si	P	S
PDMS_PA1,CN_	49.1	4.0	32.2	0.2	11.9	0.5	-
PDMS_PA1,CN-CoAM_	53.9	4.0	24.8	1.8	12.2	0.3	0.8
PDMS_PA2,CN_	35.8	2.2	39.9	0.1	21.7	0.2	-
PDMS_PA2,CN-CoAM_	54.5	5.1	24.7	1.7	-	-	1.4
PDMS_PA3,CN_	44.6	3.4	35.8	0.3	15.5	0.4	-
PDMS_PA3,CN-CoAM_	52.7	3.8	23.9	2.8	-	-	1.2
PDMS_PA5,CN_	56.2	5.1	29.8	-	8.1	0.9	-
PDMS_PA5,CN-CoAM_	54.8	5.5	24.9	1.5	10.9	0.4	1.4
PDMS_PA5_	13.9	-	55.8	-	30.3	-	-

**Table 5 materials-12-00847-t005:** Microbiological results of reduced bacterial growth by different silicone samples at different time points/after four hours of exposure.

Sample	1^st^ Day		30 Days	
	CFU/mL	Growth Reduction (%)	CFU/mL	Growth Reduction (%)
PDMS	2.64 × 10^7^	/	5.32 × 10^7^	/
PDMS_CN_	1.55 × 10^7^	41.29	5.23 × 10^7^	1.69
PDMS_CN-CoAM_	1.59 × 10^6^	**93.98**	1.35 × 10^5^	**99.75**
PDMS_PA1, CN_	3.59 × 10^7^	0.00	1.32 × 10^8^	Bacteria growth stimulation
PDMS_PA1, CN-CoAM_	1.11 × 10^7^	57.95	1.03 × 10^7^	64.80
PDMS_PA5, CN_	1.50 × 10^7^	43.18	2.52 × 10^7^	52.63
PDMS_PA5, CN-CoAM_	7.79 × 10^5^	**97.05**	2.80 × 10^7^	**67.37**

**Table 6 materials-12-00847-t006:** Microbiological results of reduced bacterial growth at different silicone samples surfaces after 1 month.

Sample	Test after One Month from Coating Application	
	CFU/mL	Growth Reduction (%)
PDMS	5.64 × 10^4^	/
PDMS_CN_	4.18 × 10^4^	25.89
PDMS_CN-CoAM_	3.92 × 10^3^	**93.05**
PDMS_PA1_, _CN_	2.55 × 10^4^	54.79
PDMS_PA1_, _CN-CoAM_	2.75 × 10^3^	**95.12**
PDMS_PA5_, _CN_	3.30 × 10^5^	Bacteria growth stimulation
PDMS_PA5_, _CN-CoAM_	1.93 × 10^3^	**96.58**

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
