# Peer review of "Functionalisation of Silicone by Drug-Embedded Chitosan Nanoparticles for Potential Applications in Otorhinolaryngology"

_materials, 2019, doi:10.3390/ma12060847_

Round 1

Reviewer 1 Report

The overall presentation is a little roundabout, sometimes too “colloquial”, with grammatical errors and typos and needs some editing to be concise and correct. I suggest the authors use grammar and spell-check options (set language to UK or US English) when they write in MS Word. Interpretation of some parts needs revision and some data not presented will need to be included.

Specific comments and corrections are given below.  

1) by reduction assay

Correction: Bacterial cell count reduction assay

2) The results show that silicone materials as representative material for tympanostomy tubes with the developed coatings are antimicrobial and as well prevent the occurrence of biofilm formation excellently.

Correction: The results show that silicone materials as suitable materials for tympanostomy tubes, with the coated developed in this study showing excellent antimicrobial and biofilm inhibition properties.

3) This implies a potential for better healing of ear inflammation, making the developed new approach for preparation of functionalized tympanostomy tubes promising for further testing towards clinical application.

Correction: This implies a potential for better healing of ear inflammation, making the newly developed approach for the preparation of functionalized tympanostomy tubes promising for further testing towards clinical applications. 

4) With the use of different medical devices, it is critical to ensure the active/functional areas of medical devices in the interface of material with the biological environment. Thus, the material surface is of a big importance; it is the main driver for material activity and functionality. Through manipulation of material surface, healing efficiency may be improved.

Correction: With the use of different medical devices, it is critical to ensure that the active/functional areas remain free of microbial contamination and the surface functionalization of the material plays a major role. Healing efficiency may be improved through the manipulation of the material surface.

5) The possibilities for its expanded use in medical applications seems unlimited, considering…..

Correction: The possibilities for its expanded use in medical applications are bright, considering….. 

6) Silicone is also a commonly used material for tympanostomy tubes, which placement is one of the most frequently performed surgical procedures in children worldwide.

Correction: Silicone is a commonly used material for tympanostomy tubes, which is a leading surgical procedure performed on children worldwide.

 7)  In addition to the mechanical function to allow fluid flow and enabling unhindered ventilation of the ear, the functionalization would also provide an antimicrobial activity through an effective and controlled drug delivery system.

Correction: …….needs to provide……controlled drug delivery.

8) Moreover, efficient drug nanostructured delivery system would synergistically improve healing

Correction: Efficient drug delivery would synergisticially…

9)  In the article [19] authors described the testing of tympanostomy tubes with a phosphoryl choline coating.

Correction: Tympanostomy tubes with a phosphoryl choline coating have been tested  [19].

10) they indicate that their sample size (n = 70) may not have been large enough to allow the coating to function effectively and thus lead to a desired improvement [19].

Correction: Sample size (n =70) may not have been large enough to allow efficient assessment of the coating and decide whether it leads to the desired improvement.

Comment: The statistics cannot change the effectiveness itself; it is the assessment of that effectiveness via which is key. 

11)   Fibronectin, a typical serum protein, which is one of the most adhesive glycoproteins, was used as a microorganism-blocking agent for tympanostomy. HSA coated tubes inhibited fibronectin binding from 59 to 85%, depending on the type of tube used. The study thus shows the potential role of HSA coating in preventing the suppression of pus and other unwanted secretions in tympanostomy tubes [20,21].

Correction: Showed the potential of HAS coating in preventing the binding of pus and other undesired secretions….

12) while the organoselenium tubes drastically inhibited biofilm formation, thus showing a potential as a long-lasting agent

Correction: …..long lasting inhibitory agent.

13) partially cytotoxic, in which they come in contact during use, as well as they may cause an allergenic reaction.

Correction: Cytotoxic, due to which they may cause an allergenic reaction by contact during use.

14) better standard living.

 Correction: better standard of living.

15) such as Cl2 (1.5 at.%),

Correction: Cl (1.5 at. %)

Comment: chlorine cannot be present as Cl2, since it is bound to the surface.

16) Among all plasma-activated samples, PDMSPA5 showed t

he highest increase of oxygen-containing groups (Fig.2, Table 4).

Correction: Showed the highest increase of ………(please delete the extra spaces).

17) SEM figure E) appears blurred (probably static/other electrical fluctuation during the measurement). The authors should provide a cleaner figure. 

 18) consisting of separate smaller particles inside, which bursted during vacuumization prior to SEM.

Correction: which burst during vacuumization

19) From the results it may also seen that antimicrobial properties are much better for those samples with encapsulated drug.

Correction: From these results, it may be seen that the encapsulated drug improves the antimicrobial properties

20) Microbiological results of reduced bacterial growth at different silicone samples surfaces after 1 months

Correction: after 1 month

21) table 6 shows that PDMSCH-TPP-CoAM, PDMSPA1, CH-TPP-CoAM and PDMSPA5, CH-TPP-CoAM have nearly the same growth reductions.

Comment: Since differences are not statistically significant, although the plasma treatment increase other desirable properties of the nanoparticles. Plasma treatment may improve the loading of nanoparticles onto the polymer, but increased hydrophilicity may encourage the adhesion of bacteria. Studies of the bacterial degradation of polymers show that increasing hydrophilicity often precedes bacterial attack.

22) Furthermore, drops or sudden jumps in concertation can indicate specific sample changes     

Correction: jumps in concentration

23) Namely, to ensure an effective therapy the bacterial cells need to be exposed to a constant antibiotic concentration in the therapeutic window for the time of exposure to a single tympanostomy tube (after this, the latter needs to be exchanged). Through this, also the body’s own immune response can be boosted.

Comment: This assertion is not accurate as the relationship between antibiotics and the host immune response is complex. According to Ankomah and Levin, www.pnas.org/cgi/doi/10.1073/pnas.1400352111

“With few exceptions, mathematical models of antibiotic treatment of patients do not consider the contribution of the host’s immune defenses. Moreover, the models of antibiotic treatment of which we are aware that do consider these defenses typically assume that the intensity of the immune response depends on the density of the infecting population of bacteria. In theory, such interaction between antibiotics and the immune response could be antagonistic rather than synergistic; by reducing the density of the bacterial population, antibiotic cidal action could decrease the intensity of the immune response that would be mobilized to eradicate the infection.”

22) Namely, regardless of the incorporated drug amount, the initially high concentration is reached in all cases, hence allows for an immediate start of the therapy after almost the same time (tonset is almost the same for all the samples).

Correction: “Onset”, not “tonset”

23)  The latter was also confirmed through an additional experiment, in which we prepared a medium with a pH of 6.0, where a higher drug concentration was achieved at all time points of the release testing.

Comment: Any data available about the pH 6.0 samples should be shown.

24) Moreover, through plasma activation, a uniform and homogeneously film coating of the applied formulations was achieved on the silicone-based substrates.

Correction: …homogenous film coating…

25) It has been found that the results of reduction of bacterial fixation on the surface of different silicone samples were better (higher percentage of decreased growth of microorganisms) with longer use of oxygen plasma prior to the adsorption of chitosan nanoparticles alone or with the embedded drug.

Correction: …bacterial adhesion…

Comment: “Fixation” describes the use of chemical fixatives to immobilize bacteria, such as during SEM sample preparation.  

26) Altogether, the results have shown that the prepared formulation have a great potential for further testing towards their potential future use in real clinical settings for tympanostomy tubes application.

Correction: have potential for future tympanostomy tubes applications, pending further testing in clinical settings. 

Author Response

Dear reviewers,

Please find below (doc attachment) our response to the comments of both reviewers regarding our manuscript. We would like to thank the reviewers for their valuable comments, which helped us to improve the manuscript. All cited issues have been clarified to the best of our knowledge. We hope that the article is now suitable for publication. The modifications to the original manuscript are tracked through changes in the revised version. In addition, please find our answers to the comments below.

We believe that these modifications improved the manuscript significantly and we hope that they will meet your expectations.

Sincerely yours, prof. dr. Lidija Fras Zemljič, (as corresponding author)

Reviewer 2 Report

This article describes the potential application of silicone functionalized by drug-embedded chitosan nanoparticles for tympanostomy tubes. The manuscript requires revision. Before the publication of the article the following issues should be addressed:

(1) In my opinion the title of the manuscript should be modified into:

 “Antimicrobial properties of silicone functionalized by drug-embedded chitosan” .

The use in the title the description “for Tympanostomy Tubes Potential Applications” suggests to the reader the application of the nanomaterial in tympanostomy tubes. It is inccorect, especially that the Authors just use it for decoration of silicone plates.

„These thin silicone plates simulated the tympanostomy tubes and due to their dimensional and geometrical characteristics, serve as ideal model platforms, allowing for an easier physicochemical characterization of functional silicone-based materials.”

(2) The „Introduction” is too long and should be markedly shortened.

(3) The system of sample decription is very complicated and makes the manuscript hard to understood.

(4) The Authors must prepare the abrreviation list.

(5) The description of Equation 1 requires correction.

“where C0 and Cs are total drug concentration used to prepare the particles and the concentration of co-amoxiclav present in the supernatant after centrifugation:.

“where C0 and Cs are total drug concentration used to prepare the particles and the concentration of co-amoxiclav present in the supernatant after centrifugation, respectively”.

(6) What masses of the samples were used during the experiment “in vitro drug release”. Fig. 5 should demonstrate the relationship: Cumulative release (%) vs. Time. It will enable the estimation of pharmaceutical availability of prepared formulation.

The kinetics of antibiotic release should be described using suitable kinetic equations (e.g. Weibull model).

Author Response

Dear reviewers,

Please find below (in doc attachment) our response to the comments of both reviewers regarding our manuscript. We would like to thank the reviewers for their valuable comments, which helped us to improve the manuscript. All cited issues have been clarified to the best of our knowledge. We hope that the article is now suitable for publication. The modifications to the original manuscript are tracked through changes in the revised version. In addition, please find our answers to the comments below.

We believe that these modifications improved the manuscript significantly and we hope that they will meet your expectations.

Sincerely yours, prof. dr. Lidija Fras Zemljič, (as corresponding author)